# Insecticidal and Detoxification Enzyme Inhibition Activities of Essential Oils for the Control of Pulse Beetle, *Callosobruchus maculatus* (F.) and *Callosobruchus chinensis* (L.) (Coleoptera: Bruchidae)

**DOI:** 10.3390/molecules28020492

**Published:** 2023-01-04

**Authors:** Himanshi Gupta, S. G. Eswara Reddy

**Affiliations:** 1Entomology Laboratory, Agrotechnology Division, CSIR-Institute of Himalayan Bioresource Technology, Palampur 176061, India; 2Academy of Scientific and Innovative Research (AcSIR), Ghaziabad 201002, India

**Keywords:** essential oils, fumigant, synergistic, repellent, ovipositional, GST, AChE

## Abstract

Pulse beetle is the most harmful pest attacking stored grains and affecting quality and marketability. Continuous use of chemical-based pesticides against pulse beetle led to the development of insecticidal resistance; essential oils (EOs) can be an effective natural alternative against this pest. The main objective was to study the chemical composition of seven EOs viz., *Acorus calamus*, *Hedychium spicatum*, *Lavandula angustifolia*, *Juniperus recurva*, *Juniperus communis*, *Cedrus deodara* and *Pinus wallichiana*, their insecticidal and enzyme inhibition activities against pulse beetle. The primary compounds present in these EOs were cis-asarone, 1,8-cineole, linalyl isobutyrate, 2-β-pinene, camphene, α-dehydro-ar-himachalene and camphene. *A. calamus* oil showed promising fumigant toxicity to *Callosobruchus maculatus* and *C. chinensis* (LC_50_ = 1357.86 and 1379.54 µL/L, respectively). A combination of *A. calamus* + *L. angustifolia* was effective against *C. maculatus* and *C. chinensis* (LC_50_ = 108.58 and 92.18 µL/L, respectively). All the combinations of EOs showed synergistic activity. In the repellency study, *A. calamus* showed more repellence to *C. maculatus* and *C. chinensis* (RC_50_ = 53.98 and 118.91 µL/L, respectively). *A. calamus* and *L. angustifolia* oil at 2500, 5000 and 10,000 µL/L significantly inhibited the AChE and GST enzymes in *C. maculatus* and *C. chinensis* after 24 and 48 h.

## 1. Introduction

The global population is continuously rising which needs a huge amount of food to feed people, so the production of food grains creates a critical source of food for growing population. Among these, pulse crops are sustainable and high-quality protein sources [1]. However, increasing pest infestation causes stored grain loss, affecting food security worldwide [2]. During storage, pulse beetle can cause a 5–10% loss in temperate and a 20–30% loss in tropical regions [3]. If no control measures are employed, there will be a 50% loss of grains in store within 3-4 months [4]. At the global level, the pulses (green/black gram) are considered as major host of pulse beetle [5] and these pulses are rich source of proteins, carbohydrates, vitamins etc. [6]. *Callosobruchus maculatus* and *Callosobruchus chinensis* are the major pest of cowpea and French beans in Africa and Asia which contains rich source of proteins (20-25%) and carbohydrates (50-60%) [7,8]. The grubs of the *C. maculatus* and *C. chinensis* feed on grains affecting quantitative and qualitative damage to the stored grains [9]. Usually, fumigation with phosphine/aluminium phosphide and chemical pesticides are more effective for controlling pulse beetle where the grains are stored in small bins and big go downs. However, fumigation causes some minor side effects on consumer’s health [10]. Also, the indiscriminate use of synthetic insecticides for the control of stored grain pests has led to insect resistance, posed harm to non-target organisms and atmosphere [11].

Several studies reported that compounds from plants substitute for synthetic insecticides and have observed that essential oils (EOs) are a better replacement for controlling stored grain pests [12,13]. Contrary to synthetic insecticides, EOs reported higher toxicity/efficacy, safer use, and more biodegradability. The higher diversity of the composition of EOs reduces the resistance development in stored pests [14,15]. Several studies have reported the insecticidal activities of botanical products of many plant species to stored grain pests [16] and these plants are predominantly used as crude/solvent extracts, powders, slurries, and volatile/crude oils [17,18]. Plants comprise of many bioactive compounds such as phenolics, flavonoids, alkaloids, tannins, sterols and EOs [19]. Plant-derived botanicals such as extracts/fractions and EOs from non-host plants with toxicant, ovipositional and repellent properties may be a better alternative for chemical pesticides to control pests [20].

The monoterpenoids and sesquiterpenoids in the EOs possess insecticidal activities against stored grain pests [21]. In earlier studies there are few reports on the insecticidal activities of *Acorus calamus* but not much literature is available for the other targeted EOs. Similarly, the mechanism of action of EOs through enzyme (Glutathione-S-Transferase and Acetylcholine esterase) inhibition activities in pulse beetle has not been reported earlier. Based on this background, the EOs of *A. calamus*, *Hedychium spicatum, Lavandula angustifolia, Juniperus recurva, Juniperus communis, Pinus wallichiana* and *Cedrus deodara* were studied for their chemical composition, fumigant, synergistic, repellent, ovipositional, and enzyme inhibition activities against pulse beetle.

## 2. Results

### 2.1. Chemical Composition of Essential Oils

The chemical composition of EOs was analyzed via GC-MS, and the chemical composition of *A. calamus* was represented in Table 1. Further, the chemical composition of remaining EOs viz *H. spicatum*, *J. recurva*, *J. communis*, *L. angustifolia*, *P. wallichiana*, and *C. deodara* were presented in Appendix A. Among different EOs, monoterpene hydrocarbons were found highest in *J. communis* (93.75%), followed by *P. wallichiana* (88.34%) and *J. recurva* (54.63%) as compared to others. In contrast, oxygenated monoterpene was found to be highest in *L. angustifolia* (75.36%), followed by *P. wallichiana* (1.99%) and *H. spicatum* (0.50%). Sesquiterpene hydrocarbon was found to be highest in *C. deodara* (76.55%), followed by *H. spicatum* (24.75%) and *J. recurva* (21.67%) as compared to others. However, oxygenated sesquiterpene was found to be highest in *H. spicatum* (3.72%), followed by *C. deodara* (12.9%) and *A. calamus* (3.93%) as compared to others. Different constituents were identified in different EOs. Among them, the major components present in the EOs of *A. calamus*, *H. spicatum*, *J. recurva*, *J. communis*, *L. angustifolia*, *P. wallichiana*, and *C. deodara* are cis-asarone (85.37%), 1,8-cineole (28.31%), 2-β-pinene (39.18%), camphene (61.03%), linalyl isobutyrate (40.16%), camphene (46.21%) and α-dehydro-ar-himachalene (44.81%), respectively as compared to other constituents.

### 2.2. Fumigant Toxicity of Essential Oils against C. maculatus and C. chinensis

Fumigant toxicity of EOs viz., *A. calamus*, *H. spicatum*, *J. recurva*, *J. communis*, *L. angustifolia*, *P. wallichiana*, and *C. deodara* were screened against adults of pulse beetle.

#### 2.2.1. *C. maculatus*

##### Toxicity of Different Essential Oils against *C. maculatus*

The fumigant toxicity of different individual essential oils (EOs) and its combinations against *C. maculatus* was presented in Table 2. Among EOs, *C. deodara* showed promising toxicity against *C. maculatus* (LC_50_ = 4116.25 µL/L) after 72 h, followed by *A. calamus* and *L. angustifolia* (LC_50_ = 4128.22 and 5204.72 µL/L, respectively) as compared to other EOs. Similarly, *A. calamus* was more effective (LC_50_ = 1357.86 µL/L) after 96 h of treatment, followed by *L. angustifolia* and *H. spicatum* (LC_50_ = 1876.15 and 2177.08 µL/L, respectively) as compared to other EOs (LC_50_ = 2818.88–6684.97 µL/L). *C. maculatus* was most susceptible to *A. calamus* after 120 h of treatment (LC_50_ = 701.48 µL/L) and was followed by *H. spicatum* and *L. angustifolia* (LC_50_ = 806.92 and 1220.93 µL/L, respectively). However, *J. recurva* was the least effective after 96 and 120 h of treatment (LC_50_ = 6684.97 and 2369.76 µL/L, respectively). All the tested EOs were superior to the positive control, i.e., aluminium phosphide (LC_50_ = 0.06 µg/mL) after 72 h.

The experimental results on a combination of different EOs *viz*., *A. calamus* + *L. angustifolia*, *A. calamus* + *H. spicatum*, *A. calamus* + *C. deodara*, *A. calamus* + *P. wallichiana*, *A. calamus* + *J. communis*, *L. angustifolia* + *H. spicatum*, *L. angustifolia* + *C. deodara*, *L. angustifolia* + *P. wallichiana* and *L. angustifolia* + *J. communis* at a 1:1 ratio against *C. maculatus* for their toxicity (LC_50_ values) and synergistic activity after 24, 48, 72 and 96 h of treatment was also presented. The EOs combinations showed toxicity against *C. maculatus* after 24 h and continued up to 96 h after treatment. Among them, *A. calamus* + *L. angustifolia* showed promising toxicity (LC_50_ = 1148.59, 533.72, and 204.01 µL/L, respectively) against *C. maculatus* after 24, 48 and 72 h, followed by *L. angustifolia* + *P. wallichiana* (LC_50_ = 1322.93, 623.70 and 312.23 µL/L) and *L. angustifolia* + *J. communis* (LC_50_ = 1376.68, 664.26 and 372.88 µL/L, respectively). Further, *A. calamus* + *L. angustifolia* showed more toxicity (LC_50_ = 108.58 µL/L) after 96 h, followed by *A. calamus* + *H. spicatum* and *A. calamus* + *P. wallichiana* (LC_50_ = 164.31 and 169.89 µL/L, respectively) as compared to other combinations. All the combinations of EOs showed synergistic activity against *C. maculatus*.

##### Percent Mortality of Essential Oils against *C. maculatus*

The fumigant toxicity of EOs, concentrations, and interaction effect (oils × concentrations) against *C. maculatus* after 72, 96 and 120 h were presented in Figure 1. The pooled mean percent mortality was found to be significantly different across the oils (F_6,174_ = 32.71, 46.13 and 32.90; *p* < 0.0001), concentrations (F_4,174_ = 181.12, 218.23 and 302.09; *p* < 0.0001) and interaction (oils × concentrations) (F_24,174_ = 2.76, 2.19 and 3.59; *p* < 0.0001) after 72, 96 and 120 h of treatment. Among oils, pooled mean percent mortality was superior in *A. calamus* (41.60 ± 19.93) after 72 h and was at par with *C. deodara* (38.80 ± 24.89) and *L. angustifolia* (34.80 ± 24.17), followed by *J. communis* (25.60 ± 14.46) as compared to other EOs. Similarly, after 96 h, pooled mean mortality was maximum in *A. calamus* (61.20 ± 21.08), followed by *L. angustifolia* (56.00 ± 24.49) compared to others. Further, after 120 h, pooled mean mortality was higher in *A. calamus* (75.20 ± 19.39) and was at par with *H. spicatum* (72.40 ± 20.67), followed by *L. angustifolia* (68.00 ± 27.23) as compared to other EOs. Among concentrations, pooled mean mortality was more at 10,000 µL/L (56.28 ± 17.16, 76.86 ± 15.,49and 94.28 ± 8.84, respectively) after 72, 96 and 120 h, as compared to other concentrations.

#### 2.2.2. *C. chinensis*

##### Toxicity of Different Essential Oils against *C. chinensis*

The fumigant toxicity of different EOs and their combinations against *C. chinensis* was presented in Table 3. Among EOs, *L. angustifolia* showed more toxicity against *C. chinensis* (LC_50_ = 4316.34 µL/L) after 72 h of treatment, followed by *C. deodara* and *J. communis* (LC_50_ = 4797.04 and 9895.41 µL/L, respectively) as compared to other oils (LC_50_ = 10,448.32–10,975.11 µL/L). Similarly, *A. calamus* (LC_50_ = 1379.54 µL/L) was more effective after 96 h and was followed by *L. angustifolia* and *H. spicatum* (LC_50_ = 1715.57 and 2598.47 µL/L, respectively) as compared to other oils (LC_50_ = 4875.82–7918.09 µL/L). However, after 120 h, *C. chinensis* was more susceptible to *L. angustifolia* (LC_50_ = 779.59 µL/L), followed by *A. calamus* and *H. spicatum* (LC_50_ = 1158.42 and 1586.15 µL/L, respectively) as compared to other oils (LC_50_ = 1694.87–2312.36 µL/L). *P. wallichiana* and *J. communis* were the least effective after 96 and 120 h (LC_50_ = 6684.97 and 2312.36 µL/L, respectively).

The experimental results on a combination of different EOs viz., *A. calamus* + *L. angustifolia*, *A. calamus* + *C. deodara*, *A. calamus* + *H. spicatum*, *A. calamus* + *J. communis*, *A. calamus* + *J. recurva*, *L. angustifolia* + *C. deodara*, *L. angustifolia* + *H. spicatum*, *L. angustifolia* + *J. communis* and *L. angustifolia* + *J. recurva* at a 1:1 ratio against *C. chinensis* for their toxicity (LC_50_ values) and synergistic activity after 24, 48, 72 and 96 h of treatment was also presented. Among combinations, *A. calamus* + *L. angustifolia* showed more toxicity after 24, 48, 72 and 96 h after treatment (LC_50_ = 396.54, 201.22, 141.89 and 92.18 µL/L, respectively) and was followed by *A. calamus* + *C. deodara* (LC_50_ = 509.92, 258.76, 182.46 and 118.54 µL/L, respectively) and *L. angustifolia* + *C. deodara* (LC_50_ = 740.11, 432.83, 279.06 and 182.66 µL/L, respectively) as compared to other combinations. All the combinations of EOs showed synergistic activity against *C. chinensis*.

##### Percent Mortality of Different EOs against *C. chinensis*

The fumigant toxicity of EOs, concentrations, and interaction effect (oils × concentrations) against *C. chinensis* after 72, 96 and 120 h were presented in Figure 2. The pooled mean percent mortality was found to be significantly different across the oils (F_6,174_ = 34.31, 51.43 and 23.72; *p* < 0.0001), concentrations (F_4,174_ = 215.79, 222.99 and 258.57; *p* < 0.0001) after 72, 96 and 120 h of treatment.

The interaction effect (oils × concentrations) showed significantly different against *C. chinensis* after 72 (F_24,174_ = 1.96; *p* < 0.0001) and 96 h (F_24,174_ = 1.89; *p* < 0.05). Results showed that among oils, pooled mean mortality was higher in *L. angustifolia* (38.00 ± 25.00) after 72 h of treatment and was at par with *C. deodara* as compared to other oils. Similarly, pooled mean mortality was maximum in *A. calamus* (62.80 ± 23.72) after 96 h and was at par with *L. angustifolia* (58.80 ± 27.43) followed by *C. deodara* (49.20 ± 21.78) as compared to EOs. After 120 h, mortality was higher in *L. angustifolia* (76.80 ± 22.49), followed by *A. calamus* (68.00 ± 24.66) compared to the remaining oils. Among concentrations, pooled mean mortality was higher in 10,000 µL/L (55.43 ± 15.40, 75.43 ± 17.71 and 92.00 ± 8.33, respectively) after 72, 96 and 120 h as compared to other concentrations.

### 2.3. Repellent Activity of Essential Oils against Pulse Beetle

The repellent activity of *A. calamus*, *H. spicatum*, *J. recurva*, *J. communis*, *L. angustifolia*, *P. wallichiana* and *C. deodara* against two species of pulse beetle after 1, 2, 3, 4, 5 and 24 h was presented in Table 4 and Table 5. The repellent activity of EOs against pulse beetle decreased over time. Results showed that *A. calamus* showed more repellence against *C. maculatus* and *C. chinensis* (RC_50_ = 53.98 and 118.91 µL/L, respectively) after 24 h and was followed by *H. spicatum* (RC_50_ = 293.77 and 226.85 µL/L, respectively) and *J. recurva* (RC_50_ = 309.75 and 383.51 µL/L, respectively) as compared to other EOs in *C. maculatus* (RC_50_ = 955.15–736.82 µL/L) and *C. chinensis* (617.11–947.27 µL/L).

### 2.4. Ovipositional Inhibition (OI) Effect of Essential Oils against Pulse Beetle

The ovipositional inhibition (OI) of different EOs and concentrations against pulse beetle reflected in the percent OI after 24, 48 and 72 h were summarized from Appendix A.

#### 2.4.1. *C. maculatus*

The percent OI of different EOs and concentrations against *C. maculatus* and its interaction effect (oils × concentrations) after 24, 48 and 72 h of treatment were presented in Appendix A. The pooled mean percent OI was significantly different across the oils (F_6,174_ = 9.38, 3.77 and 8.35; *p* < 0.0001), concentrations (F_4,174_ = 27.30, 17.78 and 30.58; *p* < 0.0001) after 24, 48 and 72 h of treatment and whereas for interaction (oils × concentrations) (F_24,174_ = 2.54 and 2.20; *p* < 0.0001) after 24 and 72 h of treatment was found significantly different, but after 48 h, the interaction was not differed significantly (*p* > 0.05). Among the EOs, the pooled mean percent OI was more in *J. communis* (33.60 ± 13.98) after 24 h, followed by *H. spicatum* and *J. recurva* (29.72 ± 11.73 and 26.28 ± 13.40, respectively) as compared to other EOs. After 48 h, pooled mean OI was maximum in *J. communis* (32.88 ± 17.71) and was at par with *C. deodara* (32.36 ± 6.80), followed by *H. spicatum* (29.40 ± 8.59) as compared to others. However, after 72 h, pooled mean percent OI was higher in *C. deodara* (36.08 ± 5.58), followed by *J. recurva* and *H. spicatum* (28.76 ± 10.56 and 28.32 ± 8.09, respectively) as compared to other remaining oils. Among concentrations, pooled mean percent OI was higher in 10,000 µL/L (36.83 ± 12.69, 37.11 ± 7.91 and 36.57 ± 7.58, respectively) after 24, 48 and 72 h as compared to other concentrations.

#### 2.4.2. *C. chinensis*

The percent OI of different EOs and concentrations against *C. chinensis* and its interaction effect (oils × concentrations) after 24, 48 and 72 h were presented in Appendix A. The pooled mean percent OI was found to be significantly different across the oils (F_6,174_ = 7.45 and 3.77; *p* < 0.0001) after 24 and 72 h, concentrations (F_4,174_ = 25.18, 20.16 and 36.46; *p* < 0.0001) after 24, 48 and 72 h of treatment. However, the mean percent OI was not differed significantly (*p* > 0.05) in the interaction (oils × concentrations). Among oils, the pooled mean OI was highest in *P. wallichiana* (34.84 ± 7.28) after 24 h, followed by *J. communis* and *C. deodara* (28.72 ± 12.05 and 27.96 ± 10.31%, respectively) as compared to other EOs. Similarly, after 48 h, pooled mean percent OI was maximum in *P. wallichiana* (32.04 ± 9.52) and was followed by *A. calamus* (27.56 ± 8.98) as compared to other EOs. In the case of 72 h, pooled mean percent OI was higher in *P. wallichiana* (32.64 ± 7.58) and was at par with *H. spicatum* (31.04 ± 7.44) followed by *A. calamus* (29.20 ± 10.61) as compared to remaining oils. Among concentrations, pooled mean percent OI was higher in 10,000 µL/L (36.03 ± 11.03, 34.05 ± 9.02 and 37.28 ± 6.45%, respectively) after 24, 48 and 72 h as compared to other concentrations.

### 2.5. Detoxification Enzyme Activities of A. Calamus and L. Angustifolia against Pulse Beetle

Detoxification enzyme inhibition activities of *A. calamus* and *L. angustifolia* against *C. maculatus* and *C. chinensis* after 24 and 48 h of treatment were presented in Figure 3a–d and Figure 4a–d. Data showed that all the concentrations of *A. calamus* inhibited the AChE activity in *C. chinensis* after 24 and 48 h (F_4,14_ = 52.42 and 50.13; *p* < 0.0001, respectively) and *C. maculatus* (F_4,14_ = 142.06 and 116.74; *p* < 0.0001, respectively), as compared to the control.

Similarly, all the concentrations of *L. angustifolia* also inhibited the AChE activity after 24 and 48 h in both *C. chinensis* (F_4,14_ = 34.15 and 67.65; *p* < 0.0001, respectively) and *C. maculatus* (F_4,14_ = 25.88 and 160.50; *p* < 0.0001, respectively), as compared to the control. Among the different concentrations evaluated, *A. calamus* at 10,000 µL/L reported higher inhibition of AChE in both *C. chinensis* and *C. maculatus* (15.64 ± 0.28 and 9.31 ± 0.99 mU/mg, respectively) and was at par with 5000 µL/L (16.07 ± 0.29 and 10.77 ± 1.30 mU/mg, respectively) after 24 h followed by other lower concentrations (1250 – 2500 µL/L). Similarly, after 48 h, *A. calamus* at 10,000 µL/L reported higher inhibition of AChE in both *C. chinensis* and *C. maculatus* (14.73 ± 0.48 and 4.31 ± 0.86 mU/mg, respectively), followed by 5000 µL/L (16.07 ± 0.29 and 8.38 ± 0.79 mU/mg, respectively) as compared to other lower concentrations (1250–2500 µL/L). *L. angustifolia* at 10,000 µL/L also showed higher inhibition of AChE after 24 and 48 h in *C. chinensis* (5.07 ± 0.76 and 3.79 ± 0.73 mU/mg, respectively) and *C. maculatus* (6.34 ± 0.13 and 3.64 ± 0.43 mU/mg, respectively), followed by 5000 µL/L in *C. chinensis* (5.65 ± 0.88 and 5.08 ± 0.10 mU/mg, respectively) and *C. maculatus* (8.25 ± 0.34 and 4.40 ± 0.15 mU/mg, respectively) as compared to other lower concentrations.

Similarly, in *C. maculatus*, all the concentrations of *A. calamus* (F_4,14_ = 20.22; *p* < 0.0001 and F_4,14_ = 42.35; *p* < 0.0001) and *L. angustifolia* (F_4,14_ = 42.09; *p* < 0.0001 and F_4,14_ = 114.40; *p* < 0.0001) significantly inhibited the GST activity after 24 and 48 h, respectively. Among concentrations, *A. calamus* at 10,000 µL/L exhibited higher inhibition (8.10 ± 2.04 and 6.84 ± 1.90 nmol/min/mL, respectively) after 24 and 48 h, followed by 5000 µL/L (23.40 ± 2.36 and 19.08 ± 2.34 nmol/min/mL, respectively), as compared to the remaining concentrations (1250-2500 µL/L). However, *L. angustifolia* at 10,000 µL/L showed more inhibition (3.52 ± 0.78 nmol/min/mL) after 24 h and was followed by 5000 µL/L (10.27 ± 0.98 nmol/min/mL), as compared to lower concentrations (1250–2500 µL/L). Likewise, after 48 h, *L. angustifolia* at 10,000 µL/L showed higher inhibition (2.39 ± 0.70 nmol/min/mL) and was at par with 5000 µL/L (5.91 ± 1.71 nmol/min/mL), as compared to other concentrations.

## 3. Discussion

The chemical composition of selected EOs, insecticidal activities (fumigant toxicity, synergistic, repellence, ovipositional activities) and their effect on detoxification enzyme inhibition in *C. chinensis* and *C. maculatus* are discussed. EOs are volatile essences and aetheroleum found in aromatic plants as a mixture of compounds produced by secondary metabolites [23]. The EOs extracted by steam/water distillation which has a lower density than water [24,25]. The composition of EO is very complex, and the individual components have valuable applications in agriculture, cosmetics, human health, and the environment. EO has been discovered to be an effective complement to synthetic compounds used in the chemical industry [26]. EOs containing monoterpenes, sesquiterpenes, and terpenoids [27] play a significant role in pesticide activities. In the present study, the chemical composition of seven EOs *viz*., *A. calamus*, *H. spicatum*, *J. recurva*, *J. communis*, *L. angustifolia*, *P. wallichiana*, and *C. deodara* were studied and presented. In the present study, the major constituents are cis-asarone in the EO of *A. calamus*; 1,8-cineole in *H. spicatum*; α-dehydro-ar-himachalene and α-himachalene in *C. deodara*; camphene in *J. communis* and *P. wallichiana*; 2-β-pinene in *J. recurva*; linalyl isobutyrate and linalool in *L. angustifolia*. The constituents of EOs play a significant role in insecticidal activities against pests. In this study, the percent composition of 1,8-cineole is higher in *H. spicatum* than in EO of *Artemisia maritima* and *M. piperita* [9,21]. Similarly, the composition of 1,8-cineole is lesser (12.2–23.15%) in *H. spicatum* oil [28,29,30] than in the current results. The composition of asarone in the present study is higher than in earlier reports, which observed comparatively less (9.5 – 50.09%) [31,32]. Earlier studies showed that the composition of linalool in *L. angustifolia* in previous studies was also lesser (19.71, 28.06, and 31.17%) [33,34,35] than present study (33.30%). In the recent study, the composition of camphene was higher (46.21%) in *P. wallichiana* as compared to the previous (0.9 – 1%) studies [36,37]. The composition of α-himachalene (18.11%) in *C. deodara* was higher in the present study than in previous studies [38]. The variation in the chemical components of EOs may be due to geographical, seasonal, and climatic conditions, genetic/hereditary, chemotype, and nutrition of the plants [39].

Insecticidal activities of the EO depend upon the presence of significant constituents, mode of application, concentration, stage, and type of insect [40,41]. In the present study, *A. calamus* showed the highest fumigant toxicity against *C. maculatus* after 96 h, compared to the previous study (LC_50_ = 3043.94 µL/L) [42]. In a similar study, *A. calamus* showed fumigant toxicity against *C. maculatus* [43] and *C. chinensis* [44,45]. The EO of *A. calamus* was less effective against *Trogaderma granarium* [46] than in present studies. *H. spicatum* is the second-best oil that showed fumigant toxicity against *C. maculatus* (LC_50_ = 2177.08 and 806.92 µL/L) and *C. chinensis* (LC_50_ = 4875.82 and 1585.15 µL/L) after 96, and 120 h of treatment and these results are comparable with that of *Plutella xylostella* [47]. Similarly, the EOs of other species *viz*., *H. forrestii*, *H. elatum*, *H. bousigonianum*, *H. flavum*, *H. thysiforme* showed promising mortality against *Stephanitis pyrioides* [48] and *H. gardnerianum* against *Artemia salina* [49]. In the present findings, the EO of *L. angustifolia* was also found effective against *C. chinensis* and *C. maculatus* (LC_50_ = 1715.57–1876.15 µL/L) after 96 h. These results agree with previous studies of *L. angustifolia* against *Ryzopertha dominica*, *Tribolium castaneum* and *L. dentata* against *C. maculatus* [50,51,52].

The binary combinations of different EOs showed promising toxicity and synergistic activity against *C. maculatus* and *C. chinensis*. Due to the non-availability of reports against pulse beetle, other EOs were compared and discussed. In the present study, among combinations *viz*., *A. calamus* + *L. angustifolia*, *A. calamus* + *H. spicatum*, and *A. calamus* + *P. wallichiana* (LC_50_ = 108.58 to 169.89 µL/L) showed more promising toxicity against *C. maculatus*. Whereas *A. calamus* + *L. angustifolia*, *A. calamus* + *C. deodara* and *L. angustifolia* + *C. deodara* (LC_50_ = 92.18 to 182.66 µL/L) also showed toxicity against *C. chinensis*. Present results are confirmed with previous studies in which the EOs of *Tagetes minuta* + *Mentha piperita* and *T. minuta* + *M. spicata* oils also showed fumigant toxicity (0.87 to 2.40 µL/mL) against *C. chinensis* and *C. maculatus* after 48 h [9]. Other researchers also reported similar findings: *Hymenocardia acida* + *Lippia adoensis* and *Lophira lanceolata* + *L. adoensis* oils showed toxicity (LC_50_ = 2.4–1.61 g kg^−1^) against *C. maculatus* [53].

All the tested EOs in the present study showed more repellence against the pulse beetle. Among them, *A. calamus* showed higher repellence against *C. maculatus* and *C. chinensis* (RC_50_ = 53.98–118.91 µL/L) and was followed by *H. spicatum* (RC_50_ = 226.85–293.77 µL/L). The present results confirmed with EOs of eucalyptus and peppermint (RC_50_ = 29.5–57.1 µL/L) against *Sitophilus oryzae* [54] and *Citrus sinensis*, *Rosmarinus officinalis* and *Pimenta racemosa* against *S. zeamais* [55].

In the current study, EOs of *C. deodara*, *J. recurva* and *A. calamus* at higher concentrations showed significant ovipositional inhibition (37.8–42.6%) against both species of pulse beetle, and these findings agree with the earlier reports in which the EOs of *C. tangerina*, *C. limonium*, *C. paradisi*, *C. aurantifolia* and *C. sinensis* [56] and *Eugenia caryophyllus* and *Illicium verum* [57] showed promising ovipositional inhibition against *C. maculatus* as compared to the present results.

AChE is categorised as the enzymes that catalyse the hydrolysis of acetylcholine (ACh), a neurotransmitter converted into acetic acid and choline [58]. GST enzyme detoxifies various insecticides, including organochlorines, pyrethroids, organophosphates, and carbamates [59]. GST activity is mainly inhibited by the chemical compounds present in the EOs [60,61]. In the present study, EOs of *A. calamus* and *L. angustifolia* were significantly inhibiting the AChE and GST activity against *C. maculatus* and *C. chinensis*. The earlier reports confirmed the current results, which reported that the EO of *A. maritima* inhibited the GST activity in *C. maculatus* [21]. Similarly, EOs of *A. monosperma*, *A. judaica*, *C. aurantifolia*, *C. viminals*, *C. lemon* and *Origanum vulgare* also inhibited the AChE and adenosine triphosphatases (ATPases) activity against *S. oryzae* [62].

## 4. Materials and Methods

### 4.1. Essential Oils (EOs)

EOs of sweet flag (*A. calamus*), spiked ginger lily (*H. spicatum*), juniper leaf (*J. recurva*), juniper berry (*J. communis*), lavender (*L. angustifolia*), pine needle (*P. wallichiana*), and cedar wood (*C. deodara*) procured from M/S Natural Biotech Products, Baggi, Mandi, Himachal Pradesh, India and extracted through steam/hydro distillation.

### 4.2. Test Insect

*C. maculatus* and *C. chinensis* obtained from Council of Scientific and Industrial Research–Central Food Technological Research Institute (CSIR–CFTRI), Mysore, Karnataka, India, for further rearing in the Entomology laboratory, CSIR-Institute of Himalayan Bioresource Technology (IHBT), Palampur, H.P, India under controlled conditions at 25 ± 2 °C temperature, 60 ± 5% relative humidity. The adults were fed on the uninfected dried green gram (*Vigna radiata* (L.)) seeds in plastic jars and covered with a black muslin cloth. The adults were inspected for growth regularly (20–30 days intervals). The newly emerged adults were then transferred to 1 L plastic jars containing uninfested seeds for mating and egg-laying to ensure sufficient adults. The adults (1–4 days old) were used for bioassay and other experiments. The dead adults removed after their adult period competed, either by sieving the grains or handpicking, depending on the number of adults.

### 4.3. Gas Chromatography (GC) Analysis

The composition of EOs determined by gas chromatography (GC) on a Shimadzu GC 2010 equipped with DB–5 (J &W Scientific, Folsom, CA, USA) fused silica capillary column (30 m × 0.25 mm, i.e., 0.25 µm film thickness) with a flame ionisation detector (FID) [21,47]. The GC oven temperature was programmed at 70 °C (initial temperature), held for 4 min, then increased at a rate of 4 °C/min to 220 °C and held for 5 min. The injector temperature was 240 °C, the detector temperature was 260 °C, and the samples were injected in split mode. The carrier gas was nitrogen at a column flow rate of 1.05 mL/min (100 kPa). The sample’s retention indices (RI) were determined based on homologous n-alkane hydrocarbons under the same conditions.

### 4.4. GC-MS Analysis and Identification

The gas chromatography-mass spectrometry (GC-MS) analysis of EOs carried out using a Shimadzu QP 2010 using a DB–5 (J&W Scientific, Folsom, CA, USA) capillary column (30 m × 0.25 mm i.d., 0.25 µm film thickness) [9]. The GC oven temperature was 70 °C for 4 min and then increased to 220 °C at 4 °C /min and held for 5 min. The injector temperature was 240 °C, the interface temperature was 250 °C, the mass acquisition range was 800–50 amu, and the ionisation energy was 70 eV. The carrier gas used was Helium. Compounds were identified using a library search of the National Institute of Standards and Technology (NIST) database [63], as well as by comparing their RI and mass spectral frame pattern with those reported in the literature [22].

### 4.5. Fumigant Toxicity of Essential Oils and Their Combinations against the Pulse Beetle

Five different test concentrations (625 to 10,000 µL/L) of EOs *viz*., *A. calamus*, *H. spicatum*, *J. recurva*, *J. communis*, *L. angustifolia*, *P. wallichiana*, and *C. deodara* and their combinations (1:1 ratio) were taken for bioassay against the adults of pulse beetle for the synergistic activity. The experiments were carried out in 15 mL glass vials. Whatman No. 9 filter paper was inserted in the inner portion of the vial cap, and EO was released into the filter papers. The vials were kept in controlled laboratory conditions to record the adult mortality at different intervals. There were five treatments per EO, and each was replicated five times. Aluminium phosphide (0.5–0.9 mg/100 g grain) was also tested as a positive control against adults of *C. maculatus*. Observations on mortality were recorded from 24 to 120 h after treatment for EOs and their combinations/binary mixtures to calculate LC_50_ values and the co-toxicity coefficient for binary mixtures.

The co-toxicity coefficient (CTC) was calculated using the formula:CTC = [LC_50_ of A/LC_50_ of A (in a mixture)] × 100(1)

If the mixture gives a CTC > 100 (synergistic action), CTC < 100 (independent action), and CTC = 100 (similar action).

### 4.6. Repellent Activity of Essential Oils against the Pulse Beetle

The repellency of EOs against pulse beetle was studied as per the suggested methodology [21,64]. Five concentrations (62.5 to 1000 µL/L) were prepared, and each concentration/treatment was replicated five times. Whatman No. 9 filter paper (diameter 9 cm) was cut and marked with a pencil into two halves, each labelled as untreated (UT) and treated (T). Filter papers were transferred to Petri plates (diameter 9 cm), treated with required concentrations of EOs, and then allowed to air dry for 5 min. Ten adults (2–3 days old) were released in the centre of the Petri plate, and the Petri plates were sealed with parafilm to prevent the escape of adults. Observations on repellency were recorded after 1, 2, 3, 4, 5, and 24 h of treatment to calculate RC_50_ values.

### 4.7. Ovipositional Deterrent Activity of Essential Oils against the Pulse Beetle

Ovipositional deterrence of EOs against pulse beetle was studied as per the suggested methodology [64]. There were five concentrations (625 to 10,000 µL/L). Five concentrations were made by mixing EOs in Tween 80. Seeds (20 no./plate) were dipped in different concentrations for 10 s, then removed and placed on filter paper to air dry for 10-15 min. Treated seeds were placed in a Petri plate (diameter 9 cm), and then ten adults of one-day old (5 male and 5 female) were released. Petri plates were sealed with parafilm to prevent the escape of the adults. For the control, seeds were treated with Tween 80 only. Each treatment was replicated five times. The number of eggs laid on seeds of green gram was observed from 24 to 72 h.

Percent oviposition inhibition was calculated by using the formula [65].
OI = [(NC − NT)/NC] ×100(2)
where NT = No. of eggs in untreated and NT = No. of eggs laid in treated.

### 4.8. Detoxification Enzyme Inhibition of A. Calamus and L. Angustifolia EO against the Pulse Beetle

#### 4.8.1. Sample Preparation

Detoxification enzymes, i.e., Acetylcholinesterase (AChE) and Glutathione-S-Transferase (GST) inhibition activities, were completed as per the standard methods [21]. Four different concentrations of *A. calamus* and *L. angustifolia* EO (10,000, 5000, 2500, and 1250 µL/L) were taken for both *C. maculatus* and *C. chinensis* for detoxification enzyme inhibition activity. The adults alive after 24 and 48 h (5–8 adults weighing 20 mg/concentration) were collected for enzyme assay. The adults in each test concentration were transferred to a centrifuge tube and homogenized phosphate buffer (pH 7.4) in a ratio of 1:9. The weight of an adult (mg): the volume of buffer (mL) was kept in a ratio of 1:9. The adults were then homogenized with a homogenizer in a micro pestle mortar. The homogenate was transferred immediately under ice bath conditions and then centrifuged at 4 °C and 12,000 rpm for 30 min. The supernatant was taken into a new centrifuge tube for protein estimation by Bradford assay [66] for all the concentrations before proceeding with enzyme assays. The same assay was repeated thrice for separate homogenates, and then average values were taken for protein estimation.

#### 4.8.2. Protein Estimation

Protein estimation was done using the Bradford method [66] by adding 2 µL of homogenate, 38 µL of MilliQ to 160 µL of Bradford reagent in triplicates. After incubation of the mixture for 15 min at room temperature, the absorbance was measured at 595 nm. Absorbance was converted into protein concentrations, and dilutions were made concerning lower concentrations for the AChE assay.

#### 4.8.3. Acetylcholinesterase Assay

The diluted 25 µL homogenates in triplicates were incubated for 30 min at room temperature with 25 µL of the reaction mixture (50 µL of DTNB, 50 µL of Acetothiocholine, and 900 µL of assay buffer). The AChE activity was spectrophotometrically measured at 410 nm in a microplate reader (Biotek, Synergy H_1_ Hybrid Multi-Mode reader) and represented as milliunits per milligram of protein (mU/mg). For the determination of AChE, the Acetylcholinesterase Assay Kit was procured from Abcam, UK.

#### 4.8.4. Glutathione-S-Transferase Assay

The reaction contains 100 µL of the solution, in which 75 µL of assay buffer, 10 µL of the homogenised sample, and 10 µL of glutathione were added. To start the reactions, 5 µL CDNB was added to each well in triplicates to the microplate at room temperature. These reaction mixtures were incubated at RT in a 96-well microplate. The enzyme kinetics were measured at the absorbance of 340 nm at 37 °C for 20 min in a microplate reader with continuous mixing for 10 s after 60 s of lag time. The extinction coefficient of 0.0096 μM−1 for CDNB was used to calculate the Glutathione-S-transferase activity and represented as nanomolar per minute per millilitre of a sample (nmol/min/mL). To determine the GST enzyme, the Glutathione-S-transferase Assay Kit was procured from Cayman Chemical, 1180 E, Ellsworth Road, Ann Arbor, MI, USA.

### 4.9. Statistical Analysis

The data on fumigant toxicity, synergistic activities, ovipositional inhibition, and repellence of different EOs was compiled. Lethal concentration (LC_50_) and repellent concentration (RC_50_) values were calculated by Probit analysis [67] using SPSS software v.16.0. The data on fumigant toxicity and ovipositional inhibition were subjected to multivariate analysis, and enzyme inhibition was subjected to one-way ANOVA by SPSS software. Means were compared by Tukey’s post hoc test to know the significant differences between treatments.

## 5. Conclusions

*A. calamus, L. angustifolia* and *H. spicatum* oil alone showed promising fumigant toxicity against both *C. maculatus* and *C. chinensis*. Among combinations, *A. calamus* oil with *L. angustifolia* and *C. deodara* against *C. chinensis* (LC_50_ = 92.18-118.54 µL/L) and *A. calamus* with *L. angustifolia* and *P. wallichiana* against *C. maculatus* (LC_50_ = 204.01-312.23 µL/L) showed promising toxicity/synergistic. *A. calamus* and *H. spicatum* also showed promising repellence to both species of pulse beetle. The insecticidal activities of promising EOs may be due to the presence of cis-asarone (85.37%), 1,8-cineole (28.31%) and 2-β-pinene (39.18%) are the major constituents. All the concentrations of *A. calamus* and *L. angustifolia* significantly inhibited the GST and AChE in both species of pulse beetle, so these enzymes may be the target site of action for tested oils in pulse beetle. Therefore, these promising EOs may be recommended to control pulse beetle where the grains are stored in bins and big godowns (staked grains) subject to large-scale trials.

## Figures and Tables

**Figure 1 molecules-28-00492-f001:**
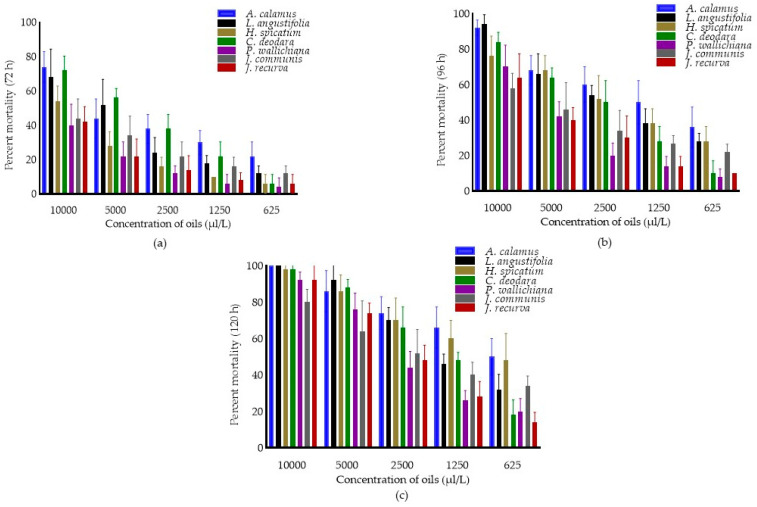
Percent mortality of essential oils in *C. maculatus* after 72 h (**a**) 96 h (**b**) and 120 h (**c**). Bars represent the standard deviation (±SD) of five replications. Means ± SD in the error bars differs significantly by Tukey’s HSD (*p* < 0.0001).

**Figure 2 molecules-28-00492-f002:**
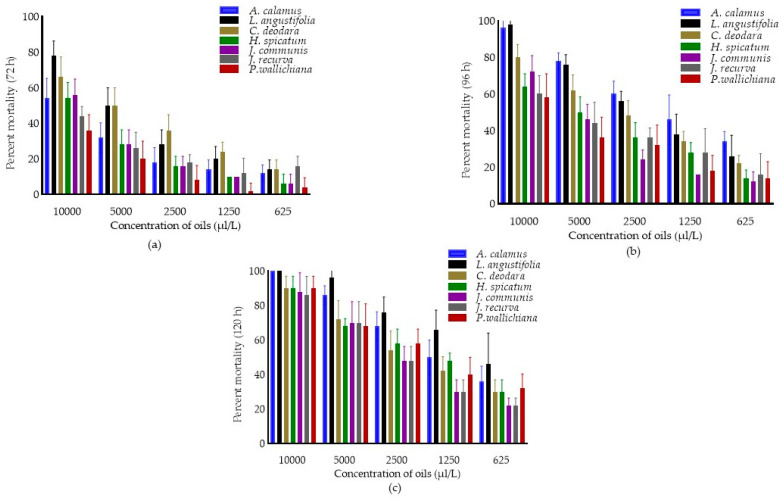
Percent mortality in *C. chinensis* after 72 h (**a**) 96 h (**b**) and 120 h (**c**). Bars represent the standard deviation (±SD) of five replications. Means ± SD in the error bars differs significantly by Tukey’s HSD (*p* < 0.0001).

**Figure 3 molecules-28-00492-f003:**
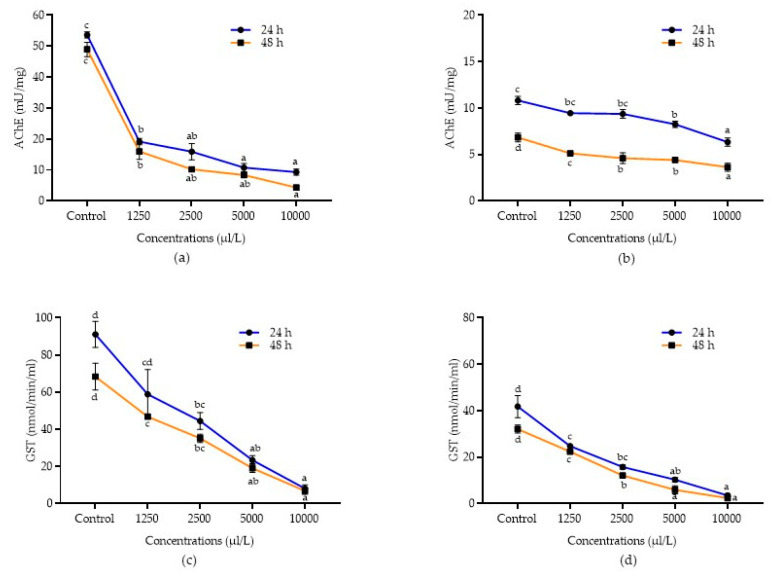
Detoxification enzyme inhibition activities of essential oils. AChE inhibition in *C. maculatus* treated with *A. calamus* (**a**) and *L. angustifolia* (**b**). GST inhibition in *C. maculatus* treated with *A. calamus* (**c**) and *L. angustifolia* (**d**). Mean (±SE) of three replications. Figures in the same letters do not differ significantly by Tukey’s HSD (*p* ≥ 0.05).

**Figure 4 molecules-28-00492-f004:**
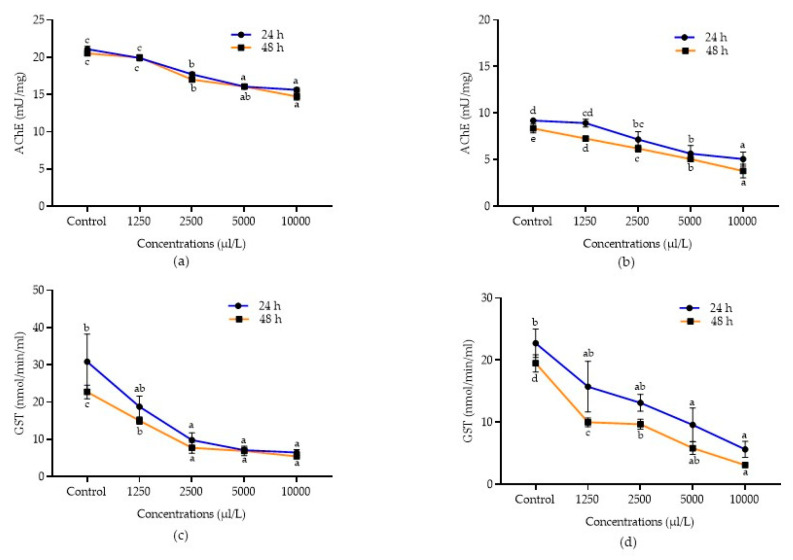
Detoxification enzyme inhibition activities of essential oils. AChE inhibition in *C. chinensis* treated with *A. calamus* (**a**) and *L. angustifolia* (**b**). GST inhibition in *C. chinensis* treated with *A. calamus* (**c**) and *L. angustifolia* (**d**). Mean (±SE) of three replications. Figures in the same letters do not differ significantly by Tukey’s HSD (*p* ≥ 0.05).

**Table 1 molecules-28-00492-t001:** Chemical composition of essential oil of *A. calamus*.

Sr. No.	Name	RI ^a^	RI ^b^	Area (%)	Mode of Identification
1	*trans–*β–Ocimene	1032	1033	0.62	MS, RI
2	β–elemene	1389	1390	0.51	MS, RI
3	Aristolene (calarene)	1431	1435	0.51	MS, RI
4	*trans* methyl isoeugenol	1451	1451	0.05	MS, RI
5	Viridiflorene	1496	1494	0.93	MS, RI
6	Bicyclo–germacrene	1500	1512	1.04	MS, RI
7	δ–cadinene	1522	1522	1.26	MS, RI
8	Kessane	1529	1528	0.76	MS, RI
9	Cedranone	1500	1504	2.93	MS, RI
10	α–calacorene	1544	1549	0.83	MS, RI
11	*cis*–asarone	1616	1617	85.37	MS, RI
12	*Trans*–asarone	1675	1686	2.64	MS, RI
13	Thujopsanone	1641	1646	0.24	MS, RI
14	Murrolene	1779	1769	0.11	MS, RI
15	Methyl palmitate	1921	1927	0.04	MS, RI
	Unidentified	–	–	1.52	
	Total			99.32	
	Monoterpene hydrocarbons *			0.67	
	Oxygenated monoterpene *			0.0	
	Phenylpropanoid			88.01	
	Sesquiterpene hydrocarbons *			5.26	
	Oxygenated sesquiterpene *			3.93	

^a^ Retention index value of compounds in the literature [22]. ^b^ Retention index value determined relative to n-alkanes (C9–C24) on the DB-5 GC column, * Percentage of compounds class in analyzed essential oil samples.

**Table 2 molecules-28-00492-t002:** Toxicity of different essential oils and their combinations against *C. maculatus*.

Essential Oils	LC_50_ (µL/L)	Confidence Limits (µL/L)	Slope ± SE	Chi-Square	*p*-Value	Co – Toxicity Coefficient	Interaction Type
*A. calamus* (72 h)	4128.22	2897.88–6763.92	1.06 ± 0.20	3.74	0.29	–	–
96 h	1357.86	879.60–1859.42	1.23 ± 0.21	3.56	0.31	–	–
120 h	701.48	401.22–986.06	1.49 ± 0.25	4.14	0.25	–	–
*H. spicatum* (72 h)	10,448.32	7116.12–20,223.58	1.41 ± 0.24	1.69	0.64	–	–
96 h	2177.08	1495.50–3067.74	1.12 ± 0.20	0.21	0.97	–	–
120 h	806.92	483.38–1116.99	1.45 ± 0.24	2.81	0.42	–	–
*J. communis* (96 h)	6497.30	4012.23–16,957.41	0.84 ± 0.20	0.41	0.94	–	–
120 h	1945.44	1256.80–2806.47	1.03 ± 0.20	0.90	0.82	–	–
*J. recurva* (96 h)	6684.97	4877.87–10,740.24	1.41 ± 0.22	1.42	0.70	–	–
120 h	2369.76	1929.69–2904.90	2.04 ± 0.23	0.78	0.85	–	–
*P. wallichiana* (96 h)	5948.84	4559.90–8576.71	1.65 ± 0.23	2.74	0.43	–	–
120 h	2313.37	1862.92–2865.74	1.92 ± 0.23	4.16	0.24	–	–
*C. deodara* (72 h)	4116.25	3240.30–5491.22	1.67 ± 0.22	0.85	0.84	–	–
96 h	2818.88	2253.91–3556.77	1.80 ± 0.22	0.79	0.85	–	–
120 h	1487.29	1202.79–1793.45	2.29 ± 0.27	1.04	0.79	–	–
*L. angustifolia* (72 h)	5204.72	3923.80–7658.14	1.46 ± 0.22	2.34	0.50	–	–
96 h	1876.15	1410.29–2419.49	1.52 ± 0.22	5.01	0.17	–	–
120 h	1220.93	961.43–1488.72	2.21 ± 0.27	3.35	0.34	–	–
AlPO_2_ (48 h)	0.075	0.068–0.085	3.85 ± 0.92	0.15	0.98	–	–
72 h	0.061	0.055–0.065	5.38 ± 0.97	2.44	0.49	–	–
96 h	0.054	0.049–0.058	6.57 ± 1.05	4.41	0.22	–	–
120 h	0.050	0.044–0.054	7.58 ± 1.24	2.94	0.40	–	–
*A. calamus* + *L. angustifolia* (24 h)	1148.59	730.42–2726.61	0.93 ± 0.20	1.51	0.68	–	–
48 h	533.72	368.26–840.46	1.00 ± 0.20	0.81	0.85	–	–
72 h	204.01	108.09–303.27	0.96 ± 0.20	2.09	0.55	2023.54	Synergistic
96 h	108.58	59.95–155.01	1.45 ± 0.24	1.68	0.64	1250.56	Synergistic
*A. calamus* + *H. spicatum* (24 h)	–	–	–	–	–	–	–
48 h	–	–	–	–	–	–	–
72 h	493.89	311.64–827.68	0.85 ± 0.19	2.31	0.51	835.86	Synergistic
96 h	164.31	77.24–250.33	0.99 ± 0.20	0.93	0.82	826.40	Synergistic
*A. calamus* + *C. deodara* (24 h)	–	–	–	–	–	–	–
48 h	–	–	–	–	–	–	–
72 h	1626.04	864.18–9816.54	0.64 ± 0.19	0.16	0.98	253.88	Synergistic
96 h	211.08	87.04 – 335.46	0.86 ± 0.20	0.35	0.95	643.29	Synergistic
*A. calamus* + *P. wallichiana* (24 h)	1967.74	1451.44–3038.81	1.34 ± 0.22	2.78	0.43	–	–
48 h	1167.83	849.94–1684.36	1.19 ± 0.20	1.31	0.73	–	–
72 h	615.31	409.99–847.10	1.18 ± 0.20	2.25	0.52	670.92	Synergistic
96 h	169.89	64.53–275.61	1.23 ± 0.24	5.20	0.16	799.26	Synergistic
*A. calamus* + *J. communis* (24 h)	–	–	–	–	–	–	–
48 h	1798.50	940.07–7815.84	0.56 ± 0.19	0.01	0.99	–	–
72 h	400.68	104.99–703.52	0.69 ± 0.19	0.06	0.99	1030.30	Synergistic
96 h	196.41	33.50–381.58	0.78 ± 0.21	1.09	0.78	691.34	Synergistic
*L. angustifolia* + *H. spicatum* (24 h)	–	–	–	–	–	–	–
48 h	1700.38	905.35–9843.58	0.67 ± 0.19	0.84	0.84	–	–
72 h	594.53	420.63–875.54	1.10 ± 0.20	1.00	0.80	875.43	Synergistic
96 h	308.73	207.86–418.60	1.25 ± 0.21	2.70	0.44	607.70	Synergistic
*L. angustifolia* + *C. deodara* (24 h)	–	–	–	–	–	–	–
48 h	1140.59	689.27–2961.81	0.73 ± 0.19	0.57	0.90	–	–
72 h	585.95	302.74–1135.58	0.66 ± 0.19	0.34	0.95	888.25	Synergistic
96 h	244.99	78.65–414.47	0.73 ± 0.19	0.89	0.83	765.81	Synergistic
*L. angustifolia* + *P. wallichiana* (24 h)	1322.93	837.00–2435.62	0.81 ± 0.19	0.15	0.98	–	–
48 h	623.70	317.43–976.71	0.82 ± 0.19	0.07	0.99	–	–
72 h	312.23	115.83–508.34	0.89 ± 0.20	0.06	0.99	1666.95	Synergistic
96 h	214.19	103.57–320.54	1.41 ± 0.25	3.99	0.26	875.93	Synergistic
*L. angustifolia* + *J. communis* (24 h)	1376.68	899.68–2310.59	0.89 ± 0.19	0.42	0.94	–	–
48 h	664.26	373.59–984.83	0.94 ± 0.20	0.32	0.96	–	–
72 h	372.88	185.63–556.04	1.11 ± 0.21	1.43	0.70	1395.82	Synergistic
96 h	251.79	126.34–371.48	1.43 ± 0.25	3.41	0.33	745.12	Synergistic

“–” LC_50_ values are not calculated where the treatment mortality was <50%; SE-Standard error; AlPO_2_-Aluminium phosphide; Slope-The change in the proportion of response of insects per unit change in dose; Chi-square- Difference between observed and expected values; *p* value-If *p*-value is greater than or equal to significance level (*p* ≥ 0.05) indicates that the observed distribution is the same as the expected distribution.

**Table 3 molecules-28-00492-t003:** Toxicity of different essential oils and their combinations against *C. chinensis*.

Essential Oils	LC_50_ (µL/L)	Confidence Limits (µL/L)	Slope ± SE	Chi-Square	*p*-Value	Co – Toxicity Coefficient	Interaction Type
*A. calamus* (72 h)	10,975.11	6875.74-27,344.73	1.10 ± 0.22	2.71	0.44	–	–
96 h	1379.54	1001.07–1778.06	1.58 ± 0.22	3.03	0.39	–	–
120 h	1158.42	863.34–1456.71	1.88 ± 0.25	3.85	0.28	–	–
*H. spicatum* (72 h)	10,448.32	7116.12–20,223.58	1.41 ± 0.24	1.69	0.64	–	–
96 h	4875.82	3464.11–8042.55	1.14 ± 0.20	0.35	0.95	–	–
120 h	1586.15	1099.21–2121.89	1.31 ± 0.21	2.28	0.52	–	–
*J. communis* (72 h)	9895.41	6865.13–18,255.56	1.45 ± 0.24	2.13	0.55	–	–
96 h	5287.16	4018.85–7673.06	1.52 ± 0.22	3.19	0.36	–	–
120 h	2312.36	1799.64–2954.15	1.63 ± 0.22	1.57	0.67	–	–
*J. recurva* (96 h)	5980.52	3945.11–12,352.29	0.97 ± 0.20	0.43	0.93	–	–
120 h	1914.97	1413.88–2500.84	1.44 ± 0.21	2.35	0.50	–	–
*P. wallichiana* (96 h)	7918.09	5152.81–17,361.77	1.05 ± 0.21	1.30	0.73	–	–
120 h	1694.87	1202.74–2250.79	1.35 ± 0.21	2.39	0.50	–	–
*C. deodara* (72 h)	4797.04	3485.81–7523.48	1.23 ± 0.21	0.04	0.99	–	–
96 h	2598.47	1923.33–3536.37	1.30 ± 0.20	0.34	0.95	–	–
120 h	1716.80	1243.41–2252.42	1.41 ± 0.21	1.62	0.65	–	–
*L. angustifolia* (72 h)	4316.34	3333.68–5798.88	1.54 ± 0.22	3.71	0.29	–	–
96 h	1715.57	1346.35–2129.90	1.86 ± 0.23	4.65	0.20	–	–
120 h	779.59	532.38–1009.98	1.95 ± 0.29	3.55	0.31	–	–
*A. calamus +* *L. angustifolia* (24 h)	396.54	256.96–618.49	0.92 ± 0.19	0.15	0.98	–	–
48 h	201.22	114.02–292.26	1.02 ± 0.20	0.62	0.89	–	–
72 h	141.89	86.94–195.48	1.39 ± 0.22	0.40	0.94	7734.94	Synergistic
96 h	92.18	55.92–124.84	1.92 ± 0.31	3.04	0.39	1496.57	Synergistic
*A. calamus* + *C. deodara* (24 h)	509.92	330.43–795.31	0.92 ± 0.19	0.15	0.98	–	–
48 h	258.76	146.62–375.82	1.02 ± 0.20	0.62	0.89	–	–
72 h	182.46	111.79–251.37	1.39 ± 0.22	0.40	0.94	6015.08	Synergistic
96 h	118.54	71.91–160.54	1.92 ± 0.31	3.04	0.39	1163.78	Synergistic
*A. calamus* + *H. spicatum* (24 h)	1196.11	828.88–1960.84	1.01 ± 0.20	0.09	0.99	–	–
48 h	455.34	298.73–623.14	1.22 ± 0.21	1.28	0.73	–	–
72 h	285.42	176.84–391.35	1.42 ± 0.23	1.38	0.71	3845.25	Synergistic
96 h	201.90	128.37–269.01	1.94 ± 0.30	0.99	0.80	683.28	Synergistic
*A. calamus* + *J. communis* (24 h)	1545.98	1116.72–2442.67	1.19 ± 0.21	0.52	0.91	–	–
48 h	911.85	648.64–1308.67	1.14 ± 0.20	0.03	0.99	–	–
72 h	512.02	337.16–703.45	1.19 ± 0.20	0.27	0.97	2143.49	Synergistic
96 h	275.64	160.70–386.58	1.37 ± 0.23	1.45	0.69	500.49	Synergistic
*A. calamus* + *J. recurva* (24 h)	2223.17	1513.69–4197.51	1.06 ± 0.20	0.11	0.99	–	–
48 h	1238.06	860.39–1941.03	1.03 ± 0.20	0.08	0.99	–	–
72 h	733.32	500.66–1020.75	1.14 ± 0.20	0.30	0.96	1496.63	Synergistic
96 h	398.87	263.00–534.18	1.43 ± 0.22	3.41	0.33	345.86	Synergistic
*L. angustifolia* + *C. deodara* (24 h)	740.11	534.33–1099.13	1.15 ± 0.20	0.29	0.96	–	–
48 h	432.83	300.38–595.76	1.19 ± 0.20	0.25	0.97	–	–
72 h	279.06	189.12–372.06	1.39 ± 0.21	0.32	0.96	1546.74	Synergistic
96 h	182.66	121.13–241.33	1.71 ± 0.25	3.03	0.39	939.21	Synergistic
*L. angustifolia* + *H. spicatum* (24 h)	1185.35	809.62–1954.79	0.97 ± 0.20	0.64	0.89	–	–
48 h	649.55	401.60–966.17	0.95 ± 0.20	0.54	0.91	–	–
72 h	354.18	159.61–550.13	0.89 ± 0.20	0.55	0.91	1218.69	Synergistic
96 h	260.18	161.91–353.63	1.58 ± 0.25	4.39	0.22	659.38	Synergistic
*L. angustifolia* + *J. communis* (24 h)	1738.22	1188.13–3148.40	1.01 ± 0.20	1.31	0.74	–	–
48 h	808.54	513.02–1235.52	0.92 ± 0.20	1.17	0.76	–	–
72 h	449.20	271.46–633.03	1.13 ± 0.20	1.26	0.74	960.89	Synergistic
96 h	284.89	183.15–381.61	1.65 ± 0.25	4.43	0.22	602.19	Synergistic
*L. angustifolia* + *J. recurva* (24 h)	2387.58	1602.20–4719.26	1.03 ± 0.20	1.65	0.65	–	–
48 h	1089.42	741.91–1663.55	1.00 ± 0.20	1.03	0.79	–	–
72 h	545.04	353.78–748.04	1.22 ± 0.21	1.85	0.60	791.93	Synergistic
96 h	259.42	18.52–498.28	1.53 ± 0.25	5.61	0.13	661.31	Synergistic

“–“LC_50_ values are not calculated where the treatment mortality was <50%; SE-Standard error.

**Table 4 molecules-28-00492-t004:** Repellent activity of different essential oils against *C. maculatus*.

Essential Oils	RC_50_ (µL/L)	Confidence Limits (µL/L)	Slope ± SE	Chi-Square	*p*-Value
*A. calamus* (1 h)	53.24	30.82–75.45	1.19 ± 0.16	3.03	0.39
2 h	17.96	3.83–37.29	0.84 ± 0.16	1.64	0.65
3 h	12.94	1.88–30.19	0.82 ± 0.17	1.63	0.65
4 h	25.24	8.50–44.79	0.98 ± 0.17	1.95	0.58
5 h	52.06	28.63–75.45	1.11 ± 0.16	4.50	0.21
24 h	53.98	20.25–89.22	0.74 ± 0.14	4.85	0.18
*H. spicatum* (1 h)	208.36	168.43–254.45	1.35 ± 0.14	4.95	0.18
2 h	109.26	84.43–134.10	1.50 ± 0.16	3.57	0.31
3 h	126.80	93.78–160.43	1.21 ± 0.15	3.53	0.32
4 h	198.92	158.10–245.46	1.27 ± 0.14	1.74	0.63
5 h	254.71	205.33–316.63	1.26 ± 0.14	2.06	0.56
24 h	293.77	248.19–350.26	1.64 ± 0.15	0.57	0.90
*J. recurva* (1 h)	176.79	140.60–216.57	1.33 ± 0.15	2.20	0.53
2 h	219.77	183.66–261.37	1.59 ± 0.15	1.40	0.71
3 h	238.35	205.39–276.16	1.95 ± 0.16	1.34	0.72
4 h	342.45	293.15–404.90	1.82 ± 0.16	3.03	0.39
5 h	359.55	304.70–431.45	1.69 ± 0.16	4.04	0.26
24 h	309.75	243.99–402.39	1.11 ± 0.14	5.03	0.16
*P. wallichiana* (1 h)	221.50	188.69–258.94	1.80 ± 0.16	3.81	0.28
2 h	269.48	234.55–310.31	2.09 ± 0.17	4.86	0.18
3 h	255.00	215.86–301.41	1.68 ± 0.15	2.56	0.46
4 h	381.69	332.91–442.18	2.17 ± 0.18	2.15	0.54
5 h	507.48	416.70–645.97	1.50 ± 0.16	1.64	0.65
24 h	955.15	686.08–1588.08	1.12 ± 0.15	1.02	0.79
*C. deodara* (1 h)	286.47	234.69–353.40	1.35 ± 0.15	1.37	0.71
2 h	389.71	313.91–503.17	1.26 ± 0.15	0.82	0.84
3 h	514.85	415.50–674.00	1.37 ± 0.15	1.13	0.77
4 h	331.02	276.27–402.95	1.51 ± 0.15	0.53	0.91
5 h	496.27	398.37–653.91	1.31 ± 0.15	0.45	0.93
24 h	425.75	335.03–572.70	1.13 ± 0.14	5.10	0.16
*L. augustifolia* (1 h)	174.41	147.28–203.85	1.82 ± 0.16	5.08	0.17
2 h	187.60	151.89–227.18	1.42 ± 0.15	3.99	0.26
3 h	228.58	187.98–276.11	1.44 ± 0.15	3.33	0.34
4 h	408.32	347.72–489.11	1.80 ± 0.16	0.37	0.95
5 h	407.11	328.26–526.27	1.27 ± 0.15	2.19	0.53
24 h	735.70	593.16–977.46	1.60 ± 0.17	2.78	0.43
*J. communis* (1 h)	213.21	144.14–302.11	0.75 ± 0.14	1.03	0.79
2 h	350.99	266.29–489.15	0.95 ± 0.14	3.70	0.30
3 h	305.05	225.62–428.66	0.87 ± 0.14	0.18	0.98
4 h	411.56	312.15–586.36	0.96 ± 0.14	0.46	0.92
5 h	356.47	253.87–548.20	0.76 ± 0.14	2.30	0.51
24 h	736.82	512.04–1327.91	0.87 ± 0.14	5.05	0.17

**Table 5 molecules-28-00492-t005:** Repellent activity of different essential oils against *C. chinensis*.

Essential Oils	RC_50_ (µL/L)	Confidence Limits (µL/L)	Slope ± SE	Chi-Square	*p*-Value
*A. calamus* (1 h)	26.04	7.99–47.60	0.89 ± 0.16	0.15	0.99
2 h	24.69	8.79–43.03	1.05 ± 0.18	0.45	0.93
3 h	12.94	1.88–30.19	0.82 ± 0.17	1.63	0.65
4 h	10.08	0.25–56.83	0.44 ± 0.14	1.74	0.63
5 h	21.67	0.16–64.21	0.40 ± 0.14	0.79	0.85
24 h	118.91	36.91–205.00	0.48 ± 0.13	0.28	0.96
*H. spicatum* (1 h)	30.60	8.86–56.30	0.79 ± 0.15	0.10	0.99
2 h	32.88	7.89–63.02	0.69 ± 0.14	0.48	0.92
3 h	44.39	7.76–87.66	0.55 ± 0.14	0.32	0.96
4 h	98.06	25.75–172.36	0.48 ± 0.13	1.25	0.74
5 h	86.14	5.75–176.71	0.35 ± 0.13	1.85	0.60
24 h	226.85	151.57–329.13	0.72 ± 0.14	4.57	0.21
*J. recurva* (1 h)	163.57	124.14–206.15	1.16 ± 0.14	1.36	0.71
2 h	216.24	174.29–265.40	1.31 ± 0.15	1.80	0.61
3 h	320.54	264.62–395.06	1.41 ± 0.15	3.29	0.35
4 h	393.61	332.45–476.02	1.68 ± 0.16	3.91	0.27
5 h	438.79	366.75–540.90	1.60 ± 0.16	0.30	0.96
24 h	383.51	313.39–484.14	1.35 ± 0.15	3.72	0.29
*P. wallichiana* (1 h)	225.56	181.99–277.35	1.30 ± 0.14	0.08	0.99
2 h	262.18	207.52–332.63	1.15 ± 0.14	3.79	0.28
3 h	363.70	291.20–470.43	1.20 ± 0.14	1.87	0.60
4 h	432.24	326.73–623.50	0.96 ± 0.14	2.41	0.49
5 h	481.32	369.81–681.46	1.06 ± 0.14	1.16	0.76
24 h	947.27	644.05–1852.50	0.90 ± 0.14	0.59	0.90
*C. deodara* (1 h)	610.61	452.44–941.18	1.00 ± 0.14	4.75	0.19
2 h	553.77	414.55–832.24	0.99 ± 0.14	0.34	0.95
3 h	629.00	487.84–888.89	1.23 ± 0.15	2.01	0.57
4 h	463.66	364.53–629.24	1.16 ± 0.14	3.48	0.32
5 h	600.86	473.37–824.52	1.28 ± 0.15	3.30	0.35
24 h	628.88	443.26–1087.98	0.84 ± 0.14	4.19	0.24
*L. augustifolia* (1 h)	113.67	88.26–139.13	1.50 ± 0.16	4.86	0.18
2 h	176.35	147.17–208.02	1.69 ± 0.16	5.20	0.16
3 h	228.58	187.98–276.11	1.44 ± 0.15	3.33	0.34
4 h	333.52	264.64–433.91	1.14 ± 0.14	5.18	0.16
5 h	392.81	306.70–531.41	1.08 ± 0.14	0.68	0.88
24 h	681.78	558.79–879.86	1.69 ± 0.17	2.19	0.53
*J. communis* (1 h)	118.35	79.08–158.04	0.98 ± 0.14	2.07	0.56
2 h	243.59	182.03–324.53	0.94 ± 0.14	0.27	0.97
3 h	334.26	253.74–461.23	0.95 ± 0.14	0.24	0.97
4 h	398.11	307.99–547.21	1.04 ± 0.14	0.63	0.89
5 h	242.53	167.66–347.66	0.76 ± 0.14	3.43	0.33
24 h	617.11	428.88–1104.77	0.80 ± 0.14	4.75	0.19

## Data Availability

The original contribution presented in the study is included in the article. Further inquiries can be directed to the corresponding author.

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
