# Peer review of "Insecticidal and Detoxification Enzyme Inhibition Activities of Essential Oils for the Control of Pulse Beetle, Callosobruchus maculatus (F.) and Callosobruchus chinensis (L.) (Coleoptera: Bruchidae)"

_molecules, 2023, doi:10.3390/molecules28020492_

Round 1
Reviewer 1 Report
The manuscript entitled "Insecticidal and detoxification enzyme inhibition activities of essential oils for the control of pulse beetle, Callosobruchus maculatus (F.) and Callosobruchus chinensis (L.) (Coleoptera: Bruchidae)" mainly evaluates the effect of seven essential oils on the insecticial and detoxification enzyme inhibition activities of pulse beetle. The methods were adequately described, and the results were clearly presented, while the novelty was insufficient. Here are some comments that helped the author improve the manuscript.
Main comments:
1. The information of the results is not readily available to the reader; Table 1-4 was suggested to be shown in figure form.
2. Moved Table S1 Chemical composition of A. calamus essential oil into manuscripts.
3. Please explan slope±SE, chi-square and p-value in the tables.
4. Please describe the novelty results or findings of the paper in the conclusion part.
Minor comments:
1. Species names are usually given in full (e.g., Acorus calamus) for the first time in the abstract, main text and figure caption, and then abbreviated (e.g., A. calamus) throughout the remainder of each part.
2. Some data lack unit in manuscript, e.g. Line 205-207;
3. There are too many titles in result part.
Reviewer 2 Report
Dear Authors,
This manuscript could not present the novelty of the work undertaken. Please redraft it so that you could pinpoint the discrete novelty statement as there are plenty of published research work (and most important it is quite very old work-Please check the attached manuscript file for more details) on the insecticidal or insect repellent properties of essential oils on post-harvest storage insect pests.
Please indicate the criteria for choice of the EO??? Also, elaborate the choice of EO for the combination experiment. The mechanism of the synergistic response of the EOs is required to be identified and specifically presented in the discussion section.
Kindly check the attached file for more specific comments.
with regards

Author Response
Point by point response to Reviewer 2 comments
Comments and Suggestions for Authors
Comment: This manuscript could not present the novelty of the work undertaken. Please redraft it so that you could pinpoint the discrete novelty statement as there are plenty of published research work (and most important it is quite very old work. Please check the attached manuscript file for more details) on the insecticidal or insect repellent properties of essential oils on post-harvest storage insect pests.
Response: Revised as suggested. In the present study, we have studied mechanism of action of two promising essential oils (A. calamus and L. angustifolia) at four different concentrations (1250, 2500, 5000 and 10000 µl/L) through enzyme inhibition activities of GST and AChE in C. chinensis and C. maculatus. All the concentrations of the both the oils significantly inhibited the GST and AChE in both species of pulse beetle. Therefore, theses enzymes may be the site of action for tested oils in pulse beetle. Similarly, A. calamus oil reported against C. chinensis but not against C. maculatus. In this study other six essential oils showed toxicity/repellence against both species of pulse beetles but earlier reports are not available for these oils against target pests.
Comment: Please indicate the criteria for choice of the EO??? Also, elaborate the choice of EO for the combination experiment. The mechanism of the synergistic response of the EOs is required to be identified and specifically presented in the discussion section.
Kindly check the attached file for more specific comments.
Response: Based on the literature and cost, the essential oils were selected and screened for their fumigant toxicity against target pest for the identification lead (s) for further future studies. Based on the fumigant toxicity of individuals oils (LC50 values), two promising oils were selected and different combinations were evaluated for their fumigant toxicity against pulse beetle. With respect to mechanism of synergy, we have not studied the mechanism of synergy of promising combinations of EOs through enzyme inhibition studies but we have studied the mechanism of action of two promising oils (A. calamus and L. angustifolia oils) in different concentrations through enzyme assay (GST and AChE), so, these enzymes may be the target site of action for tested oils in pulse beetle.
Comment: The insecticidal properties of the essential oils of the seven different plants has been carried out in several previous published reports. Here are some examples
- DOI: 10.1080/0972060X.2014.901627
- This one quite very old reference on A. calamus- EO and same test insect (C. chinensis) (10.1016/0022-474X(89)90026-X)
- Another one 10.1016/0022-474X (94)90050-R
Please ensure to identify what is the novel aspect of this work???
Response:
DOI: 10.1080/0972060X.2014.901627- In this study only Acorus calamus studied against C. chinensis but not C. maculatus. However, we have studied chemical composition of oil also. The present results are different as compared to this study. They have not calculated LC50 values and the mortality of C. chinensis was mentioned for four concentrations. In the present study, we have studied mechanism of action of two promising oils (A. calamus and L. angustifolia) through enzyme inhibition (GST and AChE) in C. chinensis and C. maculatus. This is one of the novelty of our study and these findings are not reported earlier against target pests.
10.1016/0022-474X(89)90026-X)-This is old study. Callosobruchus chinensis (L.). The EO of A. calamus was effective against C. chinensis within 48 h but not much effective against Sitophilus granarius (L.), Sitophilus ory:ae (L.), Tribolium confusum and Rhizopertha dominica (F.) which promising after week.
Another one 10.1016/0022-474X (94)90050-R-In this study, the A. calamus oil and Beta asarone was evaluated against Prostephanus truncatus (larger grain borer) but not against present target pest (pulse beetle) but their results vary which showed significant mortality after 3 weeks of treatment as compared to our findings. The results of larger grain borer cannot be compared to pulse beetle because the pulse beetle is susceptible than larger grain borer. However, fumigant toxicity of the oils varies with type of insect.
Comment: Please ensure to identify what is the novel aspect of this work???
Response: In the present study, we have studied mechanism of action of two promising essential oils (A. calamus and L. angustifolia) at four different concentrations (1250, 2500, 5000 and 10000 µl/L) through enzyme inhibition activities of GST and AChE in C. chinensis and C. maculatus. All the concentrations of the both the oils significantly inhibited the GST and AChE in both species of pulse beetle. Therefore, theses enzymes may be the site of action for tested oils in pulse beetle. Similarly, A. calamus oil reported (10.1016/0022-474X (94)90050-R)) against C. chinensis but not against C. maculatus. In this study other six essential oils showed toxicity/repellence against both species of pulse beetles but earlier reports are not available for these oils against target pests.
Comment: Synergistic insecticidal and insect repellent activities of the essential oils have also been published.
- 10.1007/s10340-021-01345-8
- 10.3390/molecules27020568
- 10.3389/fpls.2022.1016737
Identify how your results and objectives are different from the published literature.
Response:10.1007/s10340-021-01345-8-Study related to EOs against crop pest Spodoptera litura Thank you for the reference. However, in this study basil oil with other EOs studied against major crop pest (Spodoptera litura) but not against target pests of stored pulses in the current study. Therefore, the results of earlier report against S. litura may not fare to compare with pulse beetle.
10.3390/molecules27020568-Thank you for the suggestion and reference. This is my earlier study, where Mentha piperita, M. spicata and Tagetes minuta oil against pulse beetle. In this study, the combination T. minuta with Mentha spp showed synergistic activity against pulse beetle.
10.3389/fpls.2022.1016737- Thank you for the suggestion and reference. This is my earlier study, where we have studied the compounds from essential oils against mealy bug which showed toxicity and synergistic activity but these results may not have compared with the pulse beetle because sufficient literature is available related to different essential oils compounds against different type of pests (crop pests, stored grain pests).
Comment: Introduction is weak and sketchy with a poor narrative. Please improve this section.
Response: Revised as suggested.
Comment: The results of this table (Table 2) can be pooled with the EO combination tables.
Response: Thank you for the suggestion. We have merged Table 1 with Table 3 and named as Table 2 (LC50 values of oils and its combinations against C. maculatus) and Table 2 with Table 4 and named as Table 3 (LC50 values of oils and its combinations against C. chinensis).
Comment: Please rewrite the conclusion. It is quite weak.
Response: Revised as suggested.

Round 2
Reviewer 1 Report
Our questions and concerns have been answered.
Author Response
Point by point response to reviewer 1 comments
Comments and Suggestions for Authors
Our questions and concerns have been answered
Introduction and conclusion is also revised
Reviewer 2 Report
Dear Authors,
You have tried to improve the manuscript. It has more clear objectives now. But the introduction is still sketchy and not of appropriate length and narrative. There are few other suggestions for improvement of the figures. Please check the attached file.
best wishes

Author Response
Point by point response to reviewers 2 comments
Comments and suggestions for authors
You have tried to improve the manuscript. It has more clear objectives now. But the introduction is still sketchy and not of appropriate length and narrative. There are few other suggestions for improvement of the figures. Please check the attached file.
Best wishes
Response: Thank you for your valuable comments/suggestions for the improvement of the article.
The introduction is revised based on the comments and suggestions.
The grammatical and spelling errors were corrected in the revised manuscript.
Further, the figures were modified as per the reviewer’s suggestion and quality of the figures are improved.
Comment: Reviewer suggested to rephrase line 54-55 (In previous Manuscript)
Response- Thank you for the suggestion. The Introduction has been re-written, and the line has been rephrased in the revised manuscript (Line 112).
Comment- Reviewer suggested to clarify Line 63-64 in previous manuscript
Response- As per reviewer’s suggestion the ambiguity has been cleared by deleting the sentence.
Comment- Please add the lsd value or incorporate the alphabetic scripting. Kindly do this for all figures
Response- Thank you for the suggestion and revised as suggested.
Deleted portions from the manuscript
Figure 2 and Figure 4 of previous manuscript and Line 160- 165, 179-186 and 234- 246 has been deleted in the revised manuscript.
Reason- Figure 2 and figure 4 inserted in previous manuscript was not logically correct as individual essential oils combination were of five different concentrations based on LC50 values. So, nine combinations with five different concentrations (all five concentrations were different for different combinations) which makes it very difficult to express percent mortality in the figure format. So, we have deleted the figures and their respective results but given the LC50 values for all the essential oil combinations.